# Interaction of Colorectal Neoplasm Risk Factors and Association with Metabolic Health Status Focusing on Normal Waist-to-Hip Ratio in Adults

**DOI:** 10.3390/cancers16091617

**Published:** 2024-04-23

**Authors:** Ying-Chun Lin, Hung-Ju Ko, Lo-Yip Yu, Ming-Jen Chen, Horng-Yuan Wang, Shou-Chuan Shih, Chuan-Chuan Liu, Yang-Che Kuo, Kuang-Chun Hu

**Affiliations:** 1Department of Anesthesiology, MacKay Memorial Hospital, Taipei 10449, Taiwan; elegant.beaver@gmail.com; 2Graduate Institute of Epidemiology and Preventive Medicine, College of Public Health, National Taiwan University, Taipei 10449, Taiwan; 3Healthy Evaluation Center, MacKay Memorial Hospital, Taipei 10449, Taiwan; bonnie@mmh.org.tw (H.-J.K.); carrie@mmh.org.tw (C.-C.L.); 4Healthy Evaluation Center, Division of Gastroenterology, Department of Internal Medicine, MacKay Memorial Hospital, No. 92, Sec. 2, Chung-Shan North Road, Taipei 10449, Taiwan; benny7190@gmail.com (L.-Y.Y.); mmh4013@gmail.com (H.-Y.W.); shihshou@gmail.com (S.-C.S.); kuoyangche@yahoo.com.tw (Y.-C.K.); 5Division of Gastroenterology, Department of Internal Medicine, MacKay Memorial Hospital, Taipei 10449, Taiwan; mingjen.ch@gmail.com; 6MacKay Junior College of Medicine, Nursing, and Management, Taipei 10449, Taiwan

**Keywords:** waist-to-hip ratio, colorectal adenoma, Framingham Risk Score

## Abstract

**Simple Summary:**

Colorectal adenoma formation has multiple contributing factors and how these risk factors interact is not precise. Past studies have demonstrated that increased weight is one of the risk factors of colorectal adenoma formation, but does not mean participants with normal body weight are not concerned about colorectal adenoma formation. We aimed to explore the specific risk factors for colorectal adenomas in individuals with normal body weight, who are often overlooked, and the interaction between these risk factors.

**Abstract:**

Background: We aimed to evaluate the interaction between colorectal adenoma risks among asymptomatic individuals in terms of metabolic health status and obesity, and examine the normal waist-to-hip ratio (WHR) in adults with colorectal adenoma risk. Methods: A cross-sectional, retrospective study was conducted at MacKay Memorial Hospital involving 16,996 participants who underwent bidirectional gastrointestinal endoscopy between 2013 and 2023. The study recorded important clinicopathological characteristics, including age, body mass index and WHR, Framingham Risk Score (FRS), blood glucose level, and *Helicobacter pylori* (*H. pylori*) infection status. Results: Multivariate logistic regression analysis demonstrated that elevated hemoglobin A1C (HbA1c), increased FRS, positive *H. pylori* infection, and WHR ≥ 0.9 are independent risk factors for colorectal adenoma. In examining the interaction between FRS and WHR using multivariate logistic regression to evaluate adenoma risk, the OR for the interaction term was 0.95, indicating a decline in adenoma risk when considering the interaction between these two factors. Incorporating HbA1c into the analysis, evaluating the interaction between FRS and WHR still demonstrated a statistically significant impact on adenoma risk (OR 0.96, *p* < 0.001). Participants with WHR < 0.9, elevated FRS, positive *H. pylori* infection, and increased HbA1c levels were associated with a higher risk of colorectal adenoma formation. Remarkably, the increased risk of adenoma due to rising HbA1c levels was statistically significant only for those with a WHR < 0.9. Conclusions: An increase in FRS and HbA1c or a positive *H. pylori* infection still warrants vigilance for colorectal adenoma risk when WHR is 0.9. These factors interacted with each other and were found to have a minimal decline in adenoma risk when considering the interaction between WHR and FRS.

## 1. Introduction

Colorectal cancer (CRC) is the fourth most frequently diagnosed malignancy and the second leading cause of cancer-related death worldwide. In terms of sex difference, CRC is the third most common malignancy in men and the second in women [1]. Most sporadic CRC cases (approximately 85%–90%) develop from the adenoma–carcinoma sequence, and adenomas are deemed as premalignant lesions [2]. Colorectal adenoma formation has multiple contributing factors, including aging, male sex, obesity, dyslipidemia, hyperglycemia, a family history of CRC, smoking, and *Helicobacter pylori* (*H. pylori*) infection, and some of them (e.g., hemoglobin A1C (HbA1c) and *H. pylori* infection) have a synergistic effect on the risk of colorectal adenoma [3,4,5,6,7,8]. At the same time, these risk factors are also highly associated with cardiovascular disease (CVD) progression. Our previous study demonstrated that hyperglycemia combined with *H. pylori* infection could heighten the risk of synchronous carotid artery plaque and colorectal adenoma formation [9]. Although one of the factors contributing to the development of colorectal adenomas is weight gain [3], the risk remains present even in individuals with normal body weight or normal body fat. In previous studies, the interrelationship between various risk factors for colorectal adenomas has been less discussed. Colorectal adenoma formation is plagued with multiple factors [10]. Further exploration is required to understand the interactive relationships among various risk factors and their impact on this process.

This study aimed to examine potential interactions among numerous risk factors for colorectal adenomas. Furthermore, the research aims to explore the specific risk factors for colorectal adenomas in individuals with normal body weight who are often overlooked. In participants with a normal body fat percentage, we hypothesize that the increased CVD risk would also heighten colorectal adenoma risk.

## 2. Materials and Methods

### 2.1. Participant Selection

Data from 23,257 participants who had undergone annual routine checkups between January 2013 and December 2023 at MacKay Memorial Hospital, Taipei, Taiwan were retrospectively analyzed. Asymptomatic patients who underwent bidirectional endoscopy (complete colonoscopy and esophagogastroduodenoscopy [EGD]) on the same day were considered for analysis. The inclusion criteria included patients aged >20 years who underwent a screening colonoscopy and EGD along with a urease test to identify *H. pylori* infection.

The exclusion criteria were as follows: confirmed colorectal carcinoma; had a high risk of developing colorectal carcinoma, including those with inflammatory bowel syndrome, a positive family history of polypoid syndromes, or a prior history of carcinoma; inadequate bowel preparation for colonoscopy or underwent incomplete colonoscopy; lacked results for an *H. pylori* urease test on a gastric biopsy specimen or basic blood test data; or were undergoing a repeat EGD and colonoscopy as part of an annual examination. Figure 1 shows a flow chart of the study population selection.

### 2.2. Clinical Data Collection, CVD Scanning Protocol, and Questionnaire

Baseline characteristics were age, sex, height, weight, body mass index (BMI), waist circumference (WC), hip circumference (HC), waist-hip ratio (WHR), blood pressure, personal medical history and current medicine use, family history of 1st-degree relatives, and smoking habits. Data were obtained from a questionnaire completed at the time of the physical checkups. BMI was calculated as the ratio between weight (kg) and the square of height (m^2^). WC was measured in centimeters at a midpoint between the lowermost rib and the anterior superior iliac spine using a nonstretch tape. HC was measured in centimeters at the widest portion of the hip. We conducted anthropometric measurements utilizing the G-TECH GL-150 (G-Tech Co., Ltd., Gyeonggi-do, Republic of Korea), which included assessing height, weight, and body mass index. Systolic (SBP) and diastolic blood pressure (DBP) were gauged with a GE Carescape V100 Vital Signs Monitor (GE, Milwaukee, WI, USA), and the precision of blood pressure readings was assessed according to the American National Standards Institute/Association for the Advancement of Medical Instrumentation standard SP-10:1992 (with a mean error of ≤5 mm Hg and a standard deviation of ≤8 mm Hg) [11]. WHR was calculated as the ratio of WC to HC. The measurements were conducted in a laboratory accredited by TAF ISO-15189 standards [12]. Blood samples were obtained following an overnight fast (8–10 h), and levels of fasting glucose AC (mg/dL), total cholesterol (mg/dL), triglycerides (mg/dL), high-density lipoprotein (HDL), lower-density lipoprotein (LDL), and creatinine were analyzed using the UniCel DxC 800 Synchron Clinical Systems (Beckman Coulters Corporation, Brea, CA, USA). Hemoglobin A1c (HbA1c) levels were measured using a high-pressure liquid chromatography machine (Variant II, Bio-Rad, Hercules, CA, USA). Blood sample collection was obtained from the participants on the same health checkup day as when the EGDs and colonoscopy were conducted. The cardiovascular risk scores were calculated using the Framingham/Adult Treatment Panel III Risk Score (FRS), which has been validated as a predictive risk assessment tool for CVD development. The FRS was calculated separately for men and women, and the risk factors included in the FRS calculation were age, total cholesterol, HDL, systolic blood pressure, and cigarette smoking in the past 12 months [13].

### 2.3. Scanning Protocol and Definition of Helicobacter Pylori Infection and Colonic Lesions Used in Examinations

A standard esophagogastroduodenoscopy (EGD) was conducted using a gastrofibroscope (GIFQ260; Olympus Optical, Tokyo, Japan). A tissue sample was obtained from the gastric antrum using biopsy forceps and promptly tested for H. pylori using a rapid urease test (CLO, Pronto-Dry; Gastrex Corp., Warsaw, Poland) within 60 min. On the test agar, a color change from yellow to red suggests *H. pylori*, which contains intracytoplasmic urease. Experienced gastroenterologists operated the colonoscope (CF Q260AL or Q290; Olympus Optical, Tokyo, Japan), inserting it from the anus up to the ileocecal area. Large (>0.5 cm) polyps were extracted with standard polypectomy snares and small (<0.5 cm) polyps with biopsy forceps. All polyps were sent for histopathological analysis and characterized according to their macroscopic and histological results. Findings such as juvenile or inflammatory polyps, lipomas, lymphoid aggregates, and chronic nonspecific inflammation were deemed as normal mucosa. The MacKay Memorial Hospital Institutional Review Board approved this study (protocol code: 23MMHIS414e).

### 2.4. Statistical Analysis

For each participant, demographic variables were recorded. When the data for continuous variables fit a standard Gaussian distribution, a *t*-test was employed to compare participants with and without colorectal adenoma. The data are expressed as the mean ± standard deviation. A rank-sum test was employed to compare the two groups of participants when the data for continuous variables failed to fit a normal distribution. Data were expressed as the median ± interquartile range. Categorical variables were expressed as numbers (percentages), and the chi-square test was used. To calculate odds ratios (ORs) with 95% confidence intervals (CIs), univariate and multivariate logistic regression analyses were performed. Unadjusted ORs and ORs adjusted for clinically relevant predictors of adenoma were presented. Statistical significance was set at *p* < 0.05 for all variables. All analyses were performed using R version 4.1.2 (R Core Team, 2021).

## 3. Results

### 3.1. Participants’ Characteristics and Risk Factors for Colorectal Adenoma

A total of 13,564 participants (7112 males and 6452 females) were enrolled for further study. Table 1 presents differences in various physiological and health indicators between patients with and without adenomas. Specifically, individuals with adenomas were older; had a male predominance; a higher smoking rate, antihypertensive medication use, weight, BMI, WC, HC, WHR, systolic blood pressure, total cholesterol, HbA1c levels, positive Helicobacter pylori detection rate, and FRS; and a lower HDL level.

However, the average height between the two groups was nearly identical (164.89 cm vs. 164.97 cm), with a *p*-value of 0.671, suggesting that no significant difference in height was found between the presence and absence of adenomas. All other measured values had *p*-values < 0.001, signifying the statistical significance of these differences. Of note, there were no individuals with a family history of familial adenomatous polyposis (FAP) in this study.

### 3.2. Univariate and Multivariate Logistic Regression for Colorectal Adenoma Risk

Table 2 displays the outcomes derived from both univariate and multivariate logistic regression analyses examining the risk factors for colorectal adenoma. The analysis incorporates four variables: FRS, *H*. *pylori* infection status, WHR, and HbA1c. Both univariate and multivariate analyses reveal significant associations between all variables and the risk of colorectal adenoma.

In the multivariate analysis, the OR for FRS decreased from 1.09 to 1.06, the OR for *H. pylori* infection status increased from 1.44 to 1.51, the OR for WHR decreased from 2.09 to 1.40, and the OR for HbA1c decreased from 1.33 to 1.10. This suggests that the impact of these factors on colorectal adenoma risk changes after controlling for other variables. Table 2 identifies FRS, *H. pylori* infection status, and WHR as significant independent factors associated with adenoma risk. Specifically, a positive *H. pylori* infection status is strongly associated with a higher colorectal adenoma risk.

### 3.3. Evaluation of the Interaction between the Framingham Score and WHR

Multivariate logistic regression analysis was used to evaluate the interaction between FRS and WHR on adenoma risk. FRS, positive *H. pylori* infection status, WHR ≥ 0.9, and HbA1c levels were associated with an increased adenoma risk. Specifically, for every 1% increase in FRS, the adenoma risk increased by 10%; moreover, the risk increased by 50%, 78%, and 10% for a positive *H. pylori* infection status, for WHR ≥ 0.9, and for every 1% increase in HbA1c, respectively. When the interaction between FRS and WHR was analyzed, the results demonstrated that the OR for the interaction term was 0.95, suggesting a minimal decrease in adenoma risk when considering the interaction of these two factors. All of these results had *p*-values that were <0.001, indicating strong statistical significance and reliability (Table 3).

Table 3 shows the results of multivariate logistic regression analysis that assessed the interaction between FRS, HbA1c, and WHR, including their impact on adenoma risk. For every 1% increase in FRS, the adenoma risk increases by 9% (OR: 1.09, *p* < 0.001). Samples with a positive *H. pylori* test have a 1.5-fold higher adenoma risk than other samples (*p* < 0.001). Samples with a WHR ≥ 0.9 have a 4.3-fold higher adenoma risk than other samples (*p* = 0.002). Furthermore, for every one unit increase in HbA1c, the adenoma risk increases by 24% (*p* = 0.003). Although the interaction between FRS and WHR had a statistically significant impact on adenoma risk (OR 0.96, *p* < 0.001), the interaction between HbA1c and WHR did not (OR 0.85, *p* = 0.052). The likelihood ratio test also suggests accepting the smaller model, which excludes the interaction between HbA1c and WHR.

### 3.4. Multivariate Logistic Regression for the Risk of Adenoma according to WHR

FRS, *H. pylori* infection results, and HbA1c levels are associated with colorectal adenoma risk, with varying effects according to the WHR categories. The *p*-values suggest that the associations between FRS and *H. pylori* infection and the risk of adenoma are statistically significant for both WHR < 0.9 and WHR ≥ 0.9. Conversely, the association between HbA1c levels and the risk of adenoma was statistically significant for WHR < 0.9 only, but not for WHR ≥ 0.9. There was an increased risk of colorectal adenoma formation for participants with WHR < 0.9, elevated FRS, positive *H. pylori* infection, and elevated HbA1c levels (Table 4).

## 4. Discussion

Many studies have highlighted the link between being overweight and metabolic syndrome and colorectal adenoma [14,15,16,17,18]. In fact, an overweight status not only increased lower gastrointestinal but also upper gastrointestinal tract disease. Our previous study showed that a high WHR would heighten gastroesophageal reflux disease [19]. We inferred that participants with adenomas display significant differences from the healthy population in terms of age, sex, lifestyle habits, body composition indices, blood biochemistry indicators, *H. pylori* infection status, and cardiovascular risk assessment. Lee et al. showed that the high-risk group of CVD had a significantly increased risk of advanced colorectal neoplasm (OR, 3.31; 95% CI, 1.94–5.65) [20]. Chan et al. also showed that in the population undergoing coronary angiography, colorectal neoplasm prevalence was greater in patients with coronary artery disease [21]. In our data, in participants with colorectal adenoma, the FRS was significantly higher than in those without colonic lesions (6.52% vs. 3.47%, *p* < 0.001). In both groups, the FRS was below 10%, and the 10-year CVD risk was categorized as low risk. These results may contribute to a better understanding of the risk factors associated with colorectal adenomas for healthcare professionals and provide a basis for preventive and therapeutic strategies (Table 1).

In the univariate and multivariate logistic regression analyses of risk factors for colorectal adenoma, an increased FRS, WHR, and positive *H. pylori* test are significant independent factors associated with colorectal adenoma risk (Table 2). Both FRS and WHR are noninvasive examinations that are easy to perform for participants. In clinical practice, for patients with higher FRS and WHR, this could aid physicians in requesting further colonoscopy for early detection of colonic neoplasm. As FRS is a method for predicting the future 10-year CVD risk, it indicates that CVD and CRC share several similar risk factors. Both CVD and CRC are serious health issues in industrialized countries, and the study results can provide valuable insights for clinical prevention strategies.

Both WHR and FRS are usually used to assess cardiovascular risk. A higher WHR indicates more abdominal fat, which is associated with an increased cardiovascular risk. FRS, conversely, is a tool used to estimate an individual’s 10-year risk of developing CVD according to numerous factors such as age, sex, cholesterol levels, blood pressure, and smoking status [13,22,23,24]. For cardiovascular risk, to better assess cardiovascular risk, this interaction underscores the importance of not only looking at traditional risk factors included in the FRS, but also considering additional factors such as body fat distribution. Interestingly, our study shows that in colorectal adenoma, the interaction between FRS and WHR demonstrates that the OR for the interaction term was 0.95. This indicates that these two factors do not have a parallel effect on the colorectal adenoma formation, although they are independent risk factors for colonic lesions. We added HbA1c to the interaction analysis (Table 3) and found that the interaction between HbA1c and WHR had no significant impact on adenoma risk to further clarify the interaction between these colorectal adenoma risks. In our study population, a lack of predictive value for the interaction between HbA1c and WHR was suggested.

Addressing this result, the influence between smoking status and elevated HbA1c levels might play a major role. Smoking is part of FRS and is highly related to chronic disease and CVD [25]. The possibility of a mechanism might relate to chronic inflammation and gut microbiota changes. Johannsen et al. showed that smoking is also associated with increased markers of systemic vascular inflammation and C-reactive protein [26]. Lee et al. found that individuals who are presently smoking exhibit a higher percentage of the Bacteroidetes phylum, accompanied by decreased levels of Firmicutes and Proteobacteria compared to individuals who never smoked [27]. Increased Bacteroidetes spp. in the gut also positively correlated with colorectal adenoma formation [28]. Cigarette smoking habits also influence fat distribution patterns and result in a lower mean BMI compared with nonsmokers [29]. Concurrently, diabetes and obesity are also linked to a low-grade inflammation state that reflects the innate immunity activation where metabolic, environmental, and genetic factors are implicated [30]. In smoking, an elevated HbA1c and increased body fat are also linked to chronic inflammation and thus influence colorectal adenoma. Further evaluation of the inflammatory pathway and colorectal adenoma formation is warranted. Grivennikov et al. noted that when toll-like receptors were triggered by bacterial products such as endotoxins, an elevation in interleukin (IL)-23 was observed. IL-23 then affected downstream cells, including lymphocytes. Consequently, the inflammatory process was initiated, leading to subsequent increases in IL-17 and IL-6 levels. IL-17 activates the signal transducer and activator of transcription three pathways, promoting cell proliferation and survival and ultimately inducing tumorigenesis [31]. All of these risk factors could increase colon polyps through the chronic inflammatory pathway, and this could explain the interaction in colorectal adenoma formation.

According to a minimal decrease in adenoma risk when considering the interaction between FRS and WHR, we attempted to stratify participants into WHR < 0.9 and WHR ≥ 0.9 and determine the risk factors of colon neoplasm after multivariable logistic regression (Table 4). In the WHR < 0.9 group, we observed that when the FRS is increased, *H. pylori* infection positivity and an increase in HbA1c could heighten colorectal adenoma risk. In digestive system diseases, obesity is highly related to fatty liver disease, and lean body-related metabolic associated fatty liver disease has also been reported [32,33]. However, lean or normal body shape-related colorectal adenoma was elucidated less. However, this does not indicate that the body shape does not pose a risk for colorectal adenoma. Our data demonstrated that for participants who are not significantly obese but had an elevated FRS and HbA1c or have an *H. pylori* infection, colon neoplasm risk should still be considered and follow-up colon evaluation should be conducted. To the best of our knowledge, this discourse is the first to address the risk factors of colorectal adenoma in participants with normal body fat.

Our research has a few key limitations. First, this was a retrospective observational study involving patients with a relatively higher health awareness, potentially not reflecting the general population. Second, as a single medical center study, a selection bias may have been introduced. Third, owing to the nature of the cross-sectional study, it can only demonstrate an association between FRS, WHR, HbA1c, and *H. pylori* infection with the colon adenoma risk. Thus, additional cohort studies are warranted to establish a causal relationship. Despite these constraints, the large number of participants in this study helped to minimize these biases statistically.

## 5. Conclusions

An increase in WHR, FRS, and HbA1c, alongside *H. pylori* infection positivity, can heighten colorectal adenoma risk. In our study, we further evaluated the interaction between these risk factors and found a minimal decrease in adenoma risk when the interaction between WHR and FRS was considered. When the WHR is <0.9 and there is an increase in FRS and HbA1c, or an *H. pylori* infection is present, colorectal adenoma risk may still be present. These findings could enhance healthcare professionals’ comprehension of the risk factors linked to colorectal adenomas and serve as a foundation for developing preventive and therapeutic strategies.

## Figures and Tables

**Figure 1 cancers-16-01617-f001:**
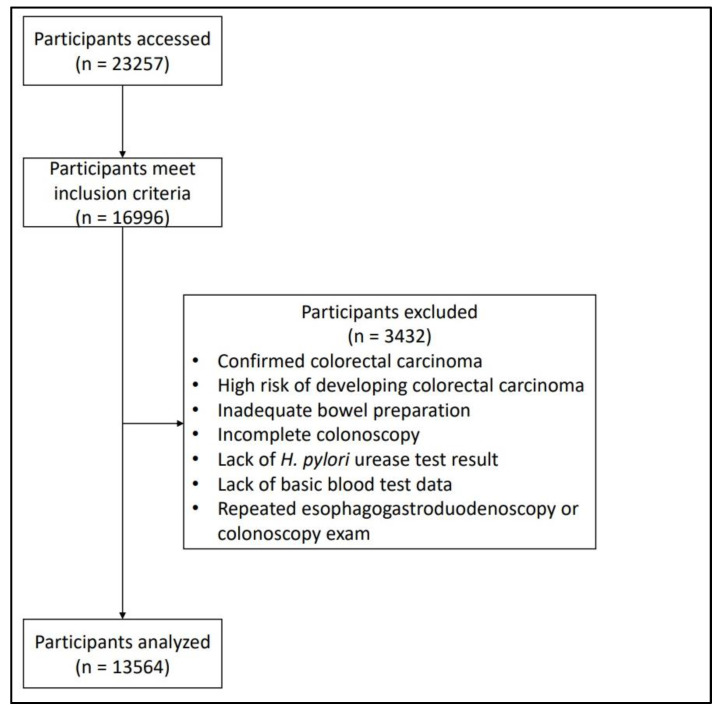
Flowchart showing selection of the study participants.

**Table 1 cancers-16-01617-t001:** Baseline demographic data characteristics (presented with mean and SD, n and %).

	Adenoma (-)		Adenoma (+)		*p*-Value
**Age (year) (mean, SD)**	46.02	11.48	53.11	10.40	<0.001
**Gender (number, %)**					<0.001
**Male (number, %)**	5289	49.76	1823	62.11	
**Female (number, %)**	5340	50.24	1112	37.89	
**Smoking (number, %)**	1443	13.57	579	19.73	<0.001
**Anti-hypertensive agent (number, %)**	1108	10.42	548	18.67	<0.001
**Body height (cm) (mean, SD)**	164.89	8.73	164.97	8.71	0.671
**Body weight (kg) (mean, SD)**	64.24	13.00	66.78	12.36	<0.001
**BMI (kg/m^2^) (mean, SD)**	23.49	3.59	24.43	3.45	<0.001
**Waist circumference (cm) (mean, SD)**	81.71	10.05	84.92	9.63	<0.001
**Buttock circumference (cm) (mean, SD)**	94.78	6.58	95.25	6.30	<0.001
**Waist-hip ratio (WHR) (mean, SD)**	0.86	0.07	0.89	0.07	<0.001
**Systolic blood pressure (mmHg) (mean, SD)**	121	17	126	18	<0.001
**Total cholesterol (mg/dL) (mean, SD)**	200.14	37.96	204.10	38.37	<0.001
**HDL (mg/dL) (mean, SD)**	54.44	15.58	52.06	15.34	<0.001
**HbA1c (%) (mean, SD)**	5.55	0.01	5.73	0.02	<0.001
***H. pylori* test positive (number, %)**	1876	30.41	754	38.57	<0.001
**Framingham score (%) (mean, SD)**	3.47	5.10	6.52	6.52	<0.001

BMI, body mass index; HbA1c, hemoglobin A1c; HDL, high-density lipoprotein; SD, standard deviation.

**Table 2 cancers-16-01617-t002:** Univariable and multivariable logistic regression for risk of colorectal adenoma.

	Univariable Analysis			Multivariable Analysis		
Variables	Odds Ratio	S.E.	*p*-Value	Odds Ratio	S.E.	*p*-Value
**Framingham score (%)**	1.09	0.00	<0.001	1.06	0.01	<0.001
***H. Pylori* infection (+)**	1.44	0.08	<0.001	1.51	0.10	<0.001
**WHR ≥ 0.9**	2.09	0.09	<0.001	1.40	0.10	<0.001
**HbA1c**	1.33	0.04	<0.001	1.10	0.04	0.009

HbA1c, hemoglobin A1c; WHR, waist-hip ratio.

**Table 3 cancers-16-01617-t003:** (**a**) Evaluating the interaction between Framingham score and WHR through multivariable logistic regression for risk of adenoma. (**b**) Evaluating the interaction between Framingham score, HbA1c and WHR through multivariable logistic regression for risk of adenoma.

	(a) **Final Model**			(b) **Exploratory Model**		
**Variables**	Odds ratio	S.E.	*p*-value	Odds ratio	S.E.	*p*-value
**Framingham score (%)**	1.10	0.01	<0.001	1.09	0.01	<0.001
***H. Pylori* infection (+)**	1.50	0.09	<0.001	1.50	0.09	<0.001
**WHR ≥ 0.9**	1.78	0.16	<0.001	4.30	1.99	0.002
**HbA1c**	1.10	0.04	0.012	1.24	0.09	0.003
**Framingham score * WHR**	0.95	0.01	<0.001	0.96	0.01	<0.001
**HbA1c * WHR**				0.85	0.07	0.052

HbA1c, hemoglobin A1c; WHR, waist-hip ratio.

**Table 4 cancers-16-01617-t004:** Multivariable logistic regression for risk of adenoma according to WHR.

	WHR < 0.9			WHR ≥ 0.9		
**Variables**	Odds Ratio	S.E.	*p*-Value	Odds Ratio	S.E.	*p*-Value
**Framingham score (%)**	1.09	0.01	<0.001	1.05	0.01	<0.001
***H. Pylori* infection (+)**	1.45	0.13	<0.001	1.55	0.14	<0.001
**HbA1c**	1.23	0.09	<0.001	1.05	0.05	0.252

HbA1c, hemoglobin A1c; WHR, waist-hip ratio.

## Data Availability

Data are contained within the article.

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
