# Peer review of "Interaction of Colorectal Neoplasm Risk Factors and Association with Metabolic Health Status Focusing on Normal Waist-to-Hip Ratio in Adults"

_cancers, 2024, doi:10.3390/cancers16091617_

Round 1

Reviewer 1 Report

Comments and Suggestions for Authors

1. The authors indicated in the Materials and Methods section that they collected the following data: personal medical history and current medicine use, family history of 1st-degree relatives, and smoking and drinking habits. However, this data is not mentioned anywhere further. Why were they collected?

2. It is interesting to look at data on the nature of nutrition and its relationship with the presence/absence of adenoma..

3. Were the adenomas single or multiple? Is there a difference in the studied indicators in this case, or does family history and heredity make a greater contribution here?

4. H. pylori infection status (+), WHR ≥ 0.9, increase in HbA1c - were these risk factors identified in different patients or could they have occurred in the same patients? If the same, then to what extent did the presence of two or more factors simultaneously increase the risk?

Author Response

Dear Reviewer,

I hope this letter finds you well. I am writing to express my sincere appreciation for your diligent review. Thank you for investing your precious time in carefully reviewing our work and providing valuable comments, leading to improvements in the current version. We would like to assure you that we have thoroughly considered each of your comments and have made every effort to address them in the current version of the paper. We hope the manuscript, after careful revisions, meet your high standards. The authors welcome further constructive comments if any.

Below we provide the point-by-point responses. All modifications in the manuscript have been highlighted (yellow) to facilitate easy identification.

Point 1.  The authors indicated in the Materials and Methods section that they collected the following data: personal medical history and current medicine use, family history of 1st-degree relatives, and smoking and drinking habits. However, this data is not mentioned anywhere further. Why were they collected?

Answer 1. Thank you for bringing this to our attention. This study collected data from individuals receiving annual routine checkups at our health evaluation center, and the listed items are information gathered from self-reported questionnaires for every participant. The analysis did not directly use personal medical history, current medicine use, and smoking but did help us calculate the Framingham risk score. Although not mentioned, a family history of first-degree relatives helps us confirm that there were no cases of familial adenomatous polyposis (FAP) which is an important risk factor for adenoma. And as you said, we did not analyze drinking habits in this article. Thus, the following changes will be made: (1) the drinking habits will be removed in this paragraph, and (2) the result of the family history of FAP will be mentioned in the results section.

Point 2. It is interesting to look at data on the nature of nutrition and its relationship with the presence/absence of adenoma.

Answer 2. Thank you for your suggestion, which could be our focus in the future. However, the nutritional effect on disease status (i.e., adenoma in this article) is very complex, and our questionnaire only collected eating habits in recent months. Thus, the data collected is not able to explore the possible causal effect, while individuals may change their eating habits according to disease status (e.g., hyperlipidemia, hyperglycemia). Furthermore, collecting eating habits and adenoma status at the same time also failed to demonstrate the possible causal time sequence. Thus, we did not analyze this. 

Point 3. Were the adenomas single or multiple? Is there a difference in the studied indicators in this case, or does family history and heredity make a greater contribution here?

Answer 3. Thank you for your suggestion. We only classified as having adenomas or not. Individuals with single or multiple adenomas are all categorized into having adenomas regardless of the numbers. Since there is no case with a family history of familial adenomatous polyposis (FAP), we did not analyze it. We will add a description in the result that no family history of FAP was noticed in our collected population.

Point 4. H. pylori infection status (+), WHR ≥ 0.9, increase in HbA1c - were these risk factors identified in different patients or could they have occurred in the same patients? If the same, then to what extent did the presence of two or more factors simultaneously increase the risk?

Answer 4. We appreciate you bringing this to our notice. This status could have occurred in the same patients. Their odds ratios to the risk of colorectal adenoma are listed in the multivariable analysis in Table 2, and the possible interactions were examined in Table 3a and Table 3b. When comparing different interaction models through likelihood ratio tests, the model with a single interaction term (i.e., Framingham score and waist-hip ratio) is suggested (i.e., result of Table 3a).

Reviewer 2 Report

Comments and Suggestions for Authors

In this study, the authors aim to analyze the potential interactions among various risk factors associated with colorectal adenomas. Additionally, the research seeks to explore the risk factors for colorectal adenomas in individuals with average body weight, an overlooked aspect. The study is exhaustive and involves many participants, which enhances its robustness. Overall, this work significantly contributes to its field, adding to the literature summarizing risk factors for colorectal cancer. However, some areas need improvement for the article to be accepted for publication:

- Introduction:

·       Improve the introduction's wording, as some ideas sound repetitive. For instance, several factors increase carcinoma formation, and Helicobacter pylori infection is mentioned twice as a factor.

·       The hypothesis suggests cardiovascular diseases will be the primary factor for adenocarcinoma formation. However, the introduction lacks sufficient background to justify the authors' hypothesis. While cardiovascular diseases are mentioned as a risk factor, it's not clear why they would pose the highest risk for individuals with average weight.

·       Define abbreviations, such as H. pylori and WHR, the first time they are used.

- Methodology:

·       It is recommended that a diagram illustrating the inclusion and exclusion criteria and the measured parameters be created.

·       The sample collection process and the conducted studies (EGD and colonoscopy) should be explained in more detail and concisely. Additionally, the authors mention measuring fasting plasma glucose, HbA1c, albumin, total cholesterol, triglycerides, low-density lipoproteins, and high-density lipoproteins (HDL) but do not describe the techniques.

·       Briefly explain how the Framingham Risk Score is calculated.

- Results:

·       While the text mentions that female subjects were also evaluated, the results for these subjects are not shown in Table 1. Furthermore, while male sex was a risk factor, this point is not easily visible due to the lack of reported results for females.

·       The wording of the results section could be improved. For example, instead of saying, “Table 2 presents the results of univariate and multivariate logistic regression analyses of the risk factors for colorectal adenoma. The four variables included were FRS, Helicobacter pylori infection status, WHR, and HbA1c. In the univariate and multivariate analyses, the results show that all variables were significantly associated with colorectal adenoma risk.” it could be rephrased as “Table 2 displays the outcomes derived from both univariate and multivariate logistic regression analyses examining the risk factors for colorectal adenoma. The analysis incorporates four variables: FRS, Helicobacter pylori infection status, WHR, and HbA1c. Both univariate and multivariate analyses reveal significant associations between all variables and the risk of colorectal adenoma.”

·       In Tables 1, 2, 3(a and b), and 4, define all abbreviations used in the table footnotes, including BMI, HDL, HbA1c, and H. pylori. Additionally, ensure consistency in using abbreviations throughout the text and tables (e.g., FRS).

·       Remove the percentage symbol where it appears next to a number in Table 1. Indicate in the data section that the numbers are percentages.

Author Response

Dear Reviewer,

I hope this letter finds you well. I'm reaching out to convey my heartfelt gratitude for your thorough evaluation. Your dedication to reviewing our work and offering insightful feedback is truly appreciated. Your input has played a pivotal role in enhancing the current version. Rest assured, we have carefully considered each of your comments and made necessary adjustments in the paper. We strive to meet your expectations with the revised manuscript. Should you have any further constructive feedback, we welcome it wholeheartedly.

Attached, you will find our detailed responses addressing each of your points. Modifications in the manuscript have been highlighted in yellow for your convenience.

Point 1.   Improve the introduction's wording, as some ideas sound repetitive. For instance, several factors increase carcinoma formation, and Helicobacter pylori infection is mentioned twice as a factor.

Answer 1. Thank you for bringing this to our attention. We improve the working in the introduction accordingly.

Point 2. The hypothesis suggests cardiovascular diseases will be the primary factor for adenocarcinoma formation. However, the introduction lacks sufficient background to justify the authors' hypothesis. While cardiovascular diseases are mentioned as a risk factor, it's not clear why they would pose the highest risk for individuals with average weight.

Answer 2. We appreciate you bringing this to our notice. In introduction section, we also mention the hyperglycemia and H. pylori infection status was co-risk factors of cardiovascular disease and colon adenoma.(reference 9) Probability underline mechanism might chronic inflammatory status of participants and we had addressed this in discussion section in line 287-292.

Point 3. Define abbreviations, such as H. pylori and WHR, the first time they are used.

Answer 3. Thanks for reminding us to add the definition for abbreviations. The definition for WHR is added in the section 2.2, and the definition of H. pylori is added in the introduction paragraph.

Point 4.  It is recommended that a diagram illustrating the inclusion and exclusion criteria and the measured parameters be created.

Answer 4. Thanks for the suggestion, a flow diagram is added in the new version in figure 1.

Point 5. The sample collection process and the conducted studies (EGD and colonoscopy) should be explained in more detail and concisely. Additionally, the authors mention measuring fasting plasma glucose, HbA1c, albumin, total cholesterol, triglycerides, low-density lipoproteins, and high-density lipoproteins (HDL) but do not describe the techniques.

Answer 5. We appreciate you bringing this to our notice. We had describe the techniques of total cholesterol, TC LDL and HDL Hba1C in section 2.2 (Beckman Coulters Corporation, USA) and (Variant II, Bio-Rad, Hercules, CA). In section 2.3 we also demonstrated more detail about EGD and colon scopy procedure in section 2.3.

Point 6. Briefly explain how the Framingham Risk Score is calculated.

Answer 6. Thank you for bringing this to our attention. The calculation of Framingham risk score is mentioned in section 2.2 as the following: “The cardiovascular risk scores were calculated using the Framingham/Adult Treatment Panel III Risk Score (FRS), which has been validated as a predictive risk assessment tool for CVD development. The FRS was calculated separately for men and women, and the risk factors included in the FRS calculation were age, total cholesterol, HDL, systolic blood pressure, and cigarette smoking in the past 12 months.”

Point 7.  While the text mentions that female subjects were also evaluated, the results for these subjects are not shown in Table 1. Furthermore, while male sex was a risk factor, this point is not easily visible due to the lack of reported results for females.

Answer 7. We add female into Table 1 under variable gender for a more clearly expression. Thanks for your suggestion.

Point 8. The wording of the results section could be improved. For example, instead of saying, “Table 2 presents the results of univariate and multivariate logistic regression analyses of the risk factors for colorectal adenoma. The four variables included were FRS, Helicobacter pylori infection status, WHR, and HbA1c. In the univariate and multivariate analyses, the results show that all variables were significantly associated with colorectal adenoma risk.” it could be rephrased as “Table 2 displays the outcomes derived from both univariate and multivariate logistic regression analyses examining the risk factors for colorectal adenoma. The analysis incorporates four variables: FRS, Helicobacter pylori infection status, WHR, and HbA1c. Both univariate and multivariate analyses reveal significant associations between all variables and the risk of colorectal adenoma.”

Answer 8. Thank you for your suggestion, the wording is rephrased as your suggestion.

Point 9. In Tables 1, 2, 3(a and b), and 4, define all abbreviations used in the table footnotes, including BMI, HDL, HbA1c, and H. pylori. Additionally, ensure consistency in using abbreviations throughout the text and tables (e.g., FRS).

Answer 9. Definitions of abbreviations are added in the table footnotes. Thanks for reminding us.

Point 10. Remove the percentage symbol where it appears next to a number in Table 1. Indicate in the data section that the numbers are percentages.

Answer 10. We appreciate you bringing this to our notice. We remove the percentage symbol next to a number in Table 1.
